# Effect of Encapsulated Nitrate and Microencapsulated Blend of Essential Oils on Growth Performance and Methane Emissions from Beef Steers Fed Backgrounding Diets

**DOI:** 10.3390/ani9010021

**Published:** 2019-01-10

**Authors:** Aklilu W. Alemu, Atmir Romero-Pérez, Rafael C. Araujo, Karen A. Beauchemin

**Affiliations:** 1Lethbridge Research and Development Centre, Agriculture and Agri-Food Canada, Lethbridge, AB T1J 4B1, Canada; aklilu.alemu@canada.ca (A.W.A.); atmir@umam.mx (A.R.-P.); 2GRASP Ind. & Com. LTDA, Curitiba, Paraná, Brazil 81260-000/EW|Nutrition GmbH, 49429 Visbek, Germany; rafael@grasp.ind.br

**Keywords:** backgrounded cattle, encapsulated nitrate, essential oil, methane

## Abstract

**Simple Summary:**

The use of supplemental dietary nitrate (NO_3_^−^) to minimize enteric methane (CH_4_) emissions from ruminants is hindered by potential toxicity effects. In the current study, the potential effects of feeding encapsulated NO_3_^−^ (EN), microencapsulated blend of essential oils (MBEO), and their combination on growth performance and enteric CH_4_ emissions of beef cattle were evaluated. There was no interaction effect between feeding EN and MBEO on CH_4_ emissions and the presence of MBEO did not affect the potential of EN to reduce CH_4_. Feeding MBEO increased CH_4_ emissions without affecting animal performance. Inclusion of EN as a replacement for urea reduced CH_4_ emissions without incurring any adverse effects on cattle health and performance.

**Abstract:**

A long-term study (112 days) was conducted to examine the effect of feeding encapsulated nitrate (NO_3_^−^), microencapsulated blend of essential oils (EO), and their combination on growth performance, feeding behavior, and enteric methane (CH_4_) emissions of beef cattle. A total of 88 crossbred steers were purchased and assigned to one of four treatments: (i) control, backgrounding high-forage diet supplemented with urea (1.17% in dietary DM); (ii) encapsulated NO_3_^−^ (EN), control diet supplemented with 2.5% encapsulated NO_3_^−^ as a replacement for urea (1.785% NO_3_^−^ in the dietary DM); (iii) microencapsulated blend of EO (MBEO), control diet supplemented with 150 mg/kg DM of microencapsulated blend of EO and pepper extract; and (iv) EN + MBEO, control diet supplemented with EN and MBEO. There was no interaction (*p* ≥ 0.080) between EN and MBEO on average dry matter intake (DMI), average daily gain (ADG), gain to feed ratio (G:F), feeding behavior, and CH_4_ emission (using GreenFeed system), implying independent effects of feeding EN and MBEO. Feeding MBEO increased CH_4_ production (165.0 versus 183.2 g/day; *p* = 0.005) and yield (18.9 versus 21.4 g/kg DMI; *p* = 0.0002) but had no effect (*p* ≥ 0.479) on average DMI, ADG, G:F, and feeding behavior. However, feeding EN had no effect on ADG and G:F (*p* ≥ 0.119) but reduced DMI (8.9 versus 8.4 kg/day; *p* = 0.003) and CH_4_ yield (21.5 versus 18.7 g/kg DMI; *p* < 0.001). Feeding EN slowed (*p* = 0.001) the feeding rate (g of DM/min) and increased (*p* = 0.002) meal frequency (events/day). Our results demonstrate that supplementing diets with a blend of EO did not lower CH_4_ emissions and there were no advantages of feeding MBEO with EN. Inclusion of EN as a replacement for urea reduced CH_4_ emissions but had no positive impact on animal performance.

## 1. Introduction

Over the past decades, livestock research has been focused on developing strategies to reduce the environmental impacts of ruminant animals [1]. As enteric methane (CH_4_) emission is the major contributor of total emissions in ruminant farming, different mitigation strategies including feed additives (e.g., inhibitors, ionophores, plant bioactive compounds, electron receptors, dietary lipids), feed (e.g., high starch grains, lipids), and feeding management (e.g., forage quality and management, feed processing, feeding frequency, precision feeding) have been directed towards minimizing enteric CH_4_ emissions [1].

Feeding nitrate (NO_3_^−^) to ruminant animals as a replacement for urea has received attention as a promising methane-mitigating approach, as several studies have shown that feeding NO_3_^−^ can decrease enteric CH_4_ [2,3,4,5,6,7]. Similarly, a recent in vitro experiment [8] and metabolism study using beef heifers [9] at our lab were also in line with the previous reports. Conversely, reduction in enteric CH_4_ was not observed from feedlot animals managed outdoors and supplemented with encapsulated NO_3_^−^ at 1.25 and 2.5% on a dry matter (DM) basis [10,11]. Furthermore, despite its positive effects on CH_4_ reduction, feeding NO_3_^−^ could pose a potential risk of NO_3_^−^/nitrite (NO_2_^−^) toxicity to animals. Nitrate intoxication can occur when the concentration of NO_2_^−^ (reduced form of NO_3_^−^) accumulates in the rumen and is absorbed into the blood stream, increasing methemoglobin (MetHb) level. When ample hemoglobin (Hb) is converted to MetHb, the animal suffers from oxygen starvation [12]. A slow release form of NO_3_^−^ (encapsulated NO_3_^−^) was developed to ensure the slow release of NO_3_^−^ to rumen microbes and minimize potential toxicity [7,8,9,10,11].

Bacterial resistance to multiple antibiotics is a worldwide health problem. As such, following the prohibition of the use of growth-promoting antibiotics in animal feeds by the European Union (1831/2003) [13], interest in the use of essential oils (EO) as potential alternatives to antibiotics and studying their effects and mechanisms on ruminal fermentation has been the focus of livestock research [14,15]. Large numbers of in vitro and in vivo studies have investigated the potential effects of EO on modifying rumen function [14,15,16]. However, the mode of action remain poorly understood [17]. Furthermore, in addition to its impact on rumen function, EO have been shown to have antioxidant, anti-inflammatory, immune modulation, mucolytic, as well as thermoregulation and blood oxygenation properties [14,18,19] and minimize stress in feedlot cattle [20]. These impacts have not been studied in detail yet. Hori et al. [21] reported that capsaicin (an alkaloid from chili pepper) increased peripheral blood flow with positive impact on body thermoregulation. Recently, Silva et al. [19,22] reported that the use of a blend of EO (Activo^®^ Premium) increased milk efficiency, digestible organic matter intake, and O_2_ saturation of Hb in dairy cows. The improved oxygenation of blood may be beneficial for cattle fed NO_3_^−^. Using the same product for sheep, Soltan et al. [23] reported a reduction in CH_4_ emissions without affecting dry matter intake (DMI) and nutrient digestibility. Overall, research on the effects of EO in beef cattle diets is fairly limited [17,24].

Therefore, the current study aimed to explore the effects of feeding encapsulated NO_3_^−^ (EN) and a microencapsulated blend of EO (MBEO) alone or in combination on feed consumption and behavior, animal performance, and enteric CH_4_ emissions from feedlot beef steers fed a high-forage diet. Because the mode of action of NO_3_^−^ and EO differ, we hypothesized that feeding NO_3_^−^ in combination with EO would improve animal performance and reduce enteric CH_4_ production.

## 2. Materials and Methods

All experimental procedures were reviewed and approved by the Animal Care and Use Committee at the Lethbridge Research and Development Centre (ACC 1626) under the guidelines of the Canadian Council on Animal Care [25] and the Veterinary Drug Directorate of Health Canada (DSTS No. 197834).

### 2.1. Animals and Experimental Design

A total of 88 crossbred steers (mean arrival BW of 287 ± 19 kg) were purchased from the local auction market. The experiment was conducted as a completely randomized design in a 2 × 2 factorial arrangement of treatments. A total of 22 animals per treatment were assigned and housed in four large adjacent pens (17 × 12.7 m; 10 m^2^ per animal). The four treatments (Table 1) were: (i) control, a typical backgrounding high-forage diet (800 g/kg DM corn silage) supplemented with urea (1.17% in dietary DM); (ii) EN, control diet supplemented with 2.5% encapsulated calcium ammonium NO_3_^−^ in dietary DM providing 1.785% NO_3_^−^ in the dietary DM (GRASP Ind. & Com. LTDA, Curitiba, Brazil); (iii) MBEO, control diet supplemented with 150 mg/kg DM of commercial microencapsulated blend of EO and pepper extract (Activo^®^ Premium, GRASP Ind. & Com. LTDA, Curitiba, Brazil); and (iv) EN + MBEO, control diet supplemented with 2.5% encapsulated calcium ammonium NO_3_^−^ in dietary DM and 150 mg/kg in the dietary DM of MBEO. The commercial blend of EO was a blend of natural and identical to natural terpenoids (carvacrol), phenylpropanoids (cinnamaldehyde and eugenol), and alkaloids (capsaicin from capsicum oleoresin) and fed to the animals according to the manufacturer’s recommended level. It was mixed with ground barley before feeding, and the blend was fed at the rate of 75 g/day to provide the full dose starting day 1. Encapsulated NO_3_^−^ was added directly into the total mixed ration (TMR) daily and contained 85.6% DM, 17.6% N, 19.6% Ca, and 71.4% NO_3_^−^ on a DM basis. Diets were formulated to be isonitrogenous, although chemical analysis indicated that the TMR containing EN were slightly lower in crude protein (CP) content (13.1 versus 14.3% DM; Table 1). The TMR were offered twice daily at 0900 h and 1600 h. Due to the high amount of Ca in encapsulated NO_3_^−^, the concentration of limestone was reduced in the EN and EN + MBEO diets to provide a similar Ca level across the diets. 

The experiment was conducted over a total of 112 days (28 days adaptation and 84 days of measurement), with the measurement period conducted in three consecutive periods of four weeks. In order to avoid the risk of intoxication, animals that received diets containing encapsulated NO_3_^−^ were acclimatized gradually using a step-up protocol during the first 28 days of adaptation; 0.625%, 1.25%, 1.875%, and 2.5% NO_3_^−^ in dietary DM. Each pen was equipped with five automated feeding stations (GrowSafe System Ltd., Airdrie, AB, Canada) to measure individual daily feed intake and feeding behavior. Animals were fitted with radio-frequency identification (RFID) ear tags to record feeding events of individual animals. Standard feedlot management procedures were implemented. Pens were bedded with straw and animals were implanted with steroids following the Standard Operating Procedure (SOP code: GEN. 1001) at Lethbridge Research and Development Centre. However, ionophores and antibiotics for liver abscess control were not added to the diets.

### 2.2. Sample Collection

Body weight (BW) was measured before feeding (nonfasted BW) on 2 consecutive days at the start and end of the experiment and once at the end of each period (4 weeks) to calculate average daily gain (ADG). Feed ingredients, TMR offered, and orts were sampled weekly and composited by period for further chemical and particle size analyses. Blood samples from all animals were collected before feeding from the jugular vein into two sodium and one lithium heparinized tubes (10-mL) and one K_2_EDTA Vacutainer^®^ tube (8-mL; Becton Dickinson Breda, Etten-Leur, The Netherland) on weeks 0 (experimental day 28, which was end of the adaptation period and beginning of Period 1), 4, 8, and 12 to determine acid-base balance, blood gases, total Hb, MetHb, packed cell volume (PCV), and NO_3_^−^ and NO_2_^−^ concentrations. Whole blood was analyzed within 30 min for total Hb and MetHb levels.

### 2.3. Emission Measurements

Methane and hydrogen were measured using the GreenFeed emission monitoring (GEM) system (C-Lock Inc., Rapid City, SD, USA). The GEM system was placed in one of the pens and animals were moved rotationally (conveyer belt approach) once a week, such that a new pen of cattle could access the system each week. Thus, once the animals were adapted to the GEM system (28 days), each treatment group had access to the system for seven days within a period, totaling three weeks of measurement per treatment group (pen) during the 84-day period. This approach allowed us to eliminate any possible pen effect because all animals spent the same amount of time in each pen. 

The GEM system allows free movement of animals (in and out of the system) and gasses are measured only when the animal’s head is in the “head chamber” unit as determined by a proximity sensor. The system is equipped with RFID reader to recognize individual animal visits by its electronic ear tag. Upon visiting the system, animals were provided with pellets from the overhead hopper (as bait) to keep them in the unit for sufficient eructation time to achieve a representative measurement. The pellet was composed of ground barley, canola meal, canola oil, dried molasses, and salt (NaCl) with a composition of 14.6% CP, 42.1% starch, 19.6% neutral detergent fiber (NDF), 11.8% acid detergent fiber (ADF), and 4.8 Mcal/kg DM gross energy (GE, DM basis). Maximum daily pellet drops per animal was set to 36 drops in the GEM system (6 visits per day × 6 drops per visit) to restrict the amount of pellet consumption. Animals could visit the system anytime during the day, but they were eligible for pellet drops only during the 6 visits. Thus, animals were required to wait for 4 h before getting their next pellet drop. The interval between pellet drops was set to 35 s to keep the animal for 3 to 7 min in the hood of the GEM system. 

Once the animal’s head was in the hood of the GEM system, air was drawn passed the nose and mouth of the animal at about 25 to 40 L/s into the collection pipe. The system measured CH_4_ and hydrogen continuously, concomitantly with air flow, temperature, atmospheric pressure, and relative humidity. Each gas was analyzed by a separate nondispersive infrared analyzer that was calibrated weekly. Daily CH_4_ emissions for individual animals were calculated by aggregating and averaging the visit flux by time of day, or “bin” over the study period, whereas hydrogen was calculated using an “arithmetic averaging method”, a straight averaging of the visit fluxes [26].

Eating behavior of the individual animal was analyzed from the GrowSafe feed bunk data. A meal was defined as a visit to the bunk, followed by an absence from the bunk for 300 s or greater. Meal size was calculated from the amount of feed consumed during a visit. Feeding rate was calculated by dividing the amount of feed consumed by meal duration (time spent at feeder), and head down duration per meal was calculated by dividing meal duration by number of meals per day. 

### 2.4. Sample Analyses

Ingredient, TMR, and ort samples were composited by period and treatment. A portion of TMR and ort samples was used to determine particle size distribution using the Penn State Particle Separator with 3 screens (18, 8, and 1.18 mm) [27]. Composited samples of ingredients, TMR, and orts were analyzed for DM content by drying at 55 °C for 72 h. Samples were ground through a 1-mm screen using a Wiley mill (A. H. Thomas, Philadelphia, PA, USA) for chemical analyses. Subsamples were further ground with a ball grinder (mixer mill MM200, Retsch, Haan, Germany) and analyzed for nitrogen (N) using flash combustion (Carlo Erba Instruments, Milan, Italy). Crude protein of ingredients was calculated by multiplying the N content by 6.25. The NDF and ADF of ingredients were determined with a FIWE 6 fiber analyzer (VELP^®^ Scientifica, Via Stazione, Italy), using the principles described by Van Soest et al. [28], including α-amylase and sodium sulfite for the NDF analysis. The GE content of ingredients and TMR was determined using a bomb calorimeter (model E2k; CAL2k, Johannesburg, South Africa). Nitrate in TMR and orts was extracted (method 968.07) [29] and the concentrations were determined using a NO_3_^−^/NO_2_^−^ Colorimetric Assay Kit (detection limit for NO_3_^−^ and NO_2_^−^ was 2 μmol/L in the original sample; Cayman Chemical Co., Ann Arbor, MI, USA). 

Blood gas and electrolytes were determined using IDEXX VetStat^®^ electrolyte and blood gas analyzer (IDEXX Laboratories, Westbrook, ME, USA). Hemoglobin and MetHb were determined using an aliquot of fresh whole blood (5 µL) from the individual animal, collected using sodium heparinized tubes (GEM OPL; Instrumentation Laboratory Company, Lexington, MA, USA). The remaining blood from sodium heparinized tube was centrifuged (AccuSpin 3/3R; Fisher Scientific, Pittsburgh, PA, USA) at 3000× *g* for 20 min at 4 °C to obtain plasma samples for NO_3_^−^ and NO_2_^−^ determination (NO_3_^−^/NO_2_^−^ Colorimetric Assay Kit; Cayman Chemical Co., Ann Arbor, MI, USA). Hematocrit samples taken in EDTA Vacutainer^®^ tubes were used to determine PCV (%) using a microcapillary reader (model MH, International Equipment Co., Boston, MA, USA). 

### 2.5. Statistical Analysis

Data were analyzed as a 2 × 2 factorial design using the MIXED procedure of SAS (SAS Inst., Inc., Cary, NC, Canada) considering animal as experimental unit. Normality of distribution and homogeneity of variance was determined using the Univariate procedure of SAS. Subsequently, data were analyzed using the following model: *y_ijk_* = *µ* + *EN_i_* + *MBEO_j_* + *EN* × *MBEO_ij_* + *e_ijk_*; where *y_ijk_* is the observation *k* in level *i* of EN and level *j* of MBEO, *µ* is the overall mean, *EN_i_* is the effect of *i*th EN treatment (control and MBEO versus EN and EN + MBEO), *MBEO_j_* is the effect of *j*th MBEO treatment (control and EN versus MBEO and EN + MBEO), *EN* × *MBEO_ij_* is the interaction of the *i*th EN and *j*th MBEO treatment, and *e_ijk_* is residual error. Period was used as a repeated measure in the model. In the case of significant interactions, the PDIFF option was included in the LSMEANS statement to account for multiple comparisons. Different time-series covariance structures were evaluated and the best one (unstructured covariance order one) was selected based on the lowest Akaike and Bayesian information criteria. Statistical significance was declared at *p* ≤ 0.05.

## 3. Results

Analyzed average NO_3_^−^ concentration in the EN diets (16.7 ± 1.10 g NO_3_^−^/kg dietary DM) was very close to the formulated level of 17.85 g NO_3_^−^/kg dietary DM. Average daily consumption of NO_3_^−^ was higher (*p* < 0.001) for EN but not affected by MBEO and EN × MBEO (Table 2). However, when daily NO_3_^−^ intake was plotted over the experimental periods, an interaction effect (*p* < 0.01) was observed where animals fed MBEO consumed more NO_3_^−^ at the beginning of the experiment (period 1) and less at the end compared with animals fed EN (Figure 1).

Final BW was reduced (*p* = 0.012) by 4.1% for EN (444 versus 463 kg) but it was not affected by MBEO (*p* = 0.336) and EN × MBEO (*p* = 0.835; Table 2). Over the experimental period, animals gained 135, 119, and 113 kg for the control, MBEO, and EN, respectively. Interaction effects between EN and MBEO on ADG and G:F were not consistent throughout the experimental period; significant interactions (*p* ≤ 0.009) were observed for day 29 to 56 and day 57 to 84 but no effect (*p* ≥ 0.20) occurred for the other days. However, the lack of interaction between EN and MBEO on average DMI (*p* = 0.479), ADG (*p* = 0.08), and average G:F (*p* = 0.240) indicates the independent effect of the two additives (Table 2). Feeding EN reduced average DMI by 6.0% (8.9 versus 8.4 for −EN and +EN, respectively; *p* = 0.003) but had no effect on ADG (*p* = 0.12) and average G:F (*p* = 0.43). However, average DMI, ADG, as well as average G:F were not affected by MBEO (*p* ≥ 0.48). 

Feeding or eating behavior was not affected (*p* ≥ 0.377) by MBEO and EN × MBEO (Table 2). However, feeding EN reduced feeding rate (g DM/min; *p* = 0.001), which resulted in longer head down duration (min/day; *p* = 0.016) during meals and greater (*p* = 0.002) meal frequency per day. 

Inclusion of EN and MBEO in the diet did not change (*p* ≥ 0.128) particle size distribution of TMR or the large (≥18 mm) and medium (8 to 18 mm) particles of orts (Table 3). However, there was an interaction (*p* < 0.04) between EN and MBEO for the small (1.2 to 8 mm) and bottom (fine, <1.2 mm) particles of orts because inclusion of EN induced selective sorting in favor of fine particles to a greater extent when MBEO was not added.

Animal visits to the GEM system and the impacts of feeding EN and MBEO on enteric CH_4_ and hydrogen emissions and yield are presented in Table 4. An interaction between EN and MBEO was observed for the number of animals that visited the GEM system (*p* = 0.021), but there were no treatment effects (*p* ≥ 0.089) on the average number of good visits or visit duration. The effect of EN and MBEO on CH_4_ emissions was independent as indicated by the lack of interaction effects (*p* ≥ 0.174) for these variables. Feeding EN reduced enteric CH_4_ emissions by 17.6% (190.9 versus 157.3 g/day for –EN and +EN, respectively; *p* < 0.0001), whereas feeding MBEO increased CH_4_ emissions by 11.0% (165.0 versus 183.2 g/day for –MBEO and +MBEO, respectively; *p* = 0.005). Similarly, CH_4_ yield (emissions corrected for intake) was 13.0% lower for EN (21.5 versus 18.7 g/kg DMI for −EN and +EN, respectively; *p* < 0.0001) but 13.6% higher for MBEO (18.9 versus 21.4 g/kg DMI for −MBEO and +MBEO, respectively; *p* = 0.0002). When CH_4_ emission was expressed in terms of energy loss, feeding EN resulted in a 10.8% lower loss of GE as CH_4_ (5.37 versus 4.79% of GE intake for −EN and +EN, respectively; *p* = 0.001), whereas feeding MBEO resulted in 10.6% more loss of GE as CH_4_ (4.83 versus 5.34% GE intake for –MBEO and +MBEO, respectively; *p* = 0.002). Correspondingly, an interaction (*p* = 0.02) was observed between additives for daily hydrogen production because addition of EN increased hydrogen production to a greater extent when MBEO was not added. However, when corrected for DMI and expressed as yield, feeding EN increased hydrogen yield by 57.3% (0.052 versus 0.081 g/kg DMI for −EN and +EN, respectively; *p* < 0.001), whereas MBEO and EN × MBEO had no effect (*p* ≥ 0.12).

The diurnal pattern of CH_4_ production and animal visits to the GEM system by hour are presented in Figure 2. Hourly CH_4_ emissions increased after morning (1000 h) and afternoon (1600 h) feeding and production rate for +EN was consistently lower throughout the day relative to −EN. Furthermore, +EN reduced CH_4_ emissions consistently throughout the experimental period, implying that the effectiveness of EN did not decline over time (Figure 3). Animals frequented the GEM system to the greatest extent at midnight (0000 h) and the lowest number of visits was observed between 0300 h and 0400 h for all treatments.

There was no interaction (*p* ≥ 0.20) between EN and MBEO on blood partial pressure of carbon dioxide (pCO_2_) and oxygen (pO_2_), total concentration of CO_2_ (tCO_2_), saturation of O_2_ (SatO_2_) and CO_2_ (SatCO_2_), bicarbonate (HCO_3_^−^), total Hb, base excess (BE), pH, and packed cell volume (PCV) (Table 5). However, an interaction between the additives was observed for blood MetHb (*p* = 0.008) and plasma NO_3_^−^-N (*p* = 0.003) contents, because blood MetHb and plasma NO_3_^−^-N increased to a lesser extent when MBEO was added. None of the animals showed visual signs of methemoglobinemia throughout the experiment, observing a maximum individual MetHb concentration of 4.1% of total Hb for the +EN +MBEO treatment, a level that is not a threat to animal health and wellbeing. Furthermore, feeding EN (+EN) increased (*p* = 0.05) blood HCO_3_^−^ and total CO_2_ relative to treatment without EN. Feeding MBEO had no effect on all measured blood parameters. Plasma NO_3_^−^-N concentration for the EN and EN + MBEO treatments was reduced over the experimental period (Figure 4a); however, plasma NO_2_^−^-N concentrations reached maximum at the fourth week of the experimental period for all the treatments and sharply reduced thereafter (Figure 4b).

## 4. Discussion

Supplementation of NO_3_^−^ in ruminant diets has been proposed as an alternative to increase non-protein nitrogen intake while effectively minimizing enteric CH_4_ emissions [30]. However, it is also well documented that over-consumption of NO_3_^−^ can be toxic to animals [12]. Encapsulation of NO_3_^−^ has been used [7,29] to ensure slow release of NO_3_^−^ in the rumen and increase the efficiency of microbes to fully reduce NO_3_^−^ to ammonia, thus minimizing the risk of NO_3_^−^/NO_2_^−^ toxicity. 

Essential oils have been shown to favorably affect rumen fermentation in vitro, but the observed responses have not translated into improved production characteristics in the few existing studies with beef cattle [17]. Furthermore, previous studies have reported that the immune modulation, antioxidant, thermoregulation, and blood oxygenation properties of EO may improve animal productivity and energetics [14,18]. Despite the use of NO_3_^−^ and EO in ruminant diets, previous studies have not explored their possible interaction on enteric CH_4_ mitigation and animal productivity. The main finding in the current study is that the effects of EN were mostly independent from those of MBEO, as most of the variables examined showed a lack of significant interaction between EN and MBEO, thus considering the responses to these additives as generally independent. 

### 4.1. Nitrate

In the literature, the impact of feeding NO_3_^−^ to ruminants on DMI varies among studies. For example, using unencapsulated NO_3_^−^ (2% in diet DM) Lund et al. [31] reported 11% reduction in DMI for dairy cattle fed a high-forage diet (58% DM). Similarly, Hulshof et al. [2] used unencapsulated NO_3_^−^ (2.2% in dietary DM) and observed a decrease in DMI of 6% in beef cattle fed high-forage diets (60% DM). Encapsulation of NO_3_^−^ ensures not only slow release in the rumen [8] but also has a potential to minimize its negative impact on feed intake caused by its organoleptic properties [32]. This has been the case in some previous reports that used encapsulated NO_3_^−^ [7,10] but not in others [11]. 

In our study, the lack of effects of EN on ADG and G:F were in agreement with Lee et al. [10] and El-Zaiat et al. [7]. Lee et al. [10] supplemented encapsulated NO_3_^−^ (2.5% in dietary DM) to beef cattle fed a high-forage diet (65% DM corn silage) and reported no effect on DMI and feed efficiency. Whereas with the same inclusion rate in a high concentrate diet (80% DM of barley grain), the same authors observed a 7.5% reduction in DMI and 11% improvement in feed efficiency for finishing beef cattle [11]. Changes in feeding and eating behavior following NO_3_^−^ supplementation may contribute to DMI and feed efficiency responses [32,33]. For example, Lee et al. [10] observed significant sorting of the TMR for large particles, which increased the proportion of small particles and decreased the proportion of large particles in orts, as well as a considerable increase in NO_3_^−^ concentration in orts. Conversely, in our study, feeding EN induced sorting in favor of fine particles but had no effect on either the large and medium particles of orts or the total amount of orts (% of total offered). The reduction in feeding rate (g DM/min) for EN was manifested in longer head down duration during meals and more frequent meals per day. These changes in feeding behavior of cattle fed EN were consistent with Velazco et al. [34], where NO_3_^−^-fed cattle consumed a large number of meals per day and smaller in size when compared to cattle fed a control diet. 

Previous studies reported a reduction in enteric CH_4_ production in several species and categories of animals due to NO_3_^−^ feeding [1,30]. Reductions in CH_4_ yield ranging between 4% (with 1% NO_3_^−^ in diet DM; [9,10]) and 33% (with 2.7% NO_3_^−^ in diet DM; [7]) have been reported for ruminants fed high-forage diets supplemented with encapsulated NO_3_^−^. The observed reduction in CH_4_ yield in our study for +EN (13.0%) was within the range of 12.2% and 18.3% reduction reported for beef heifers fed a forage diet (50% DM barley silage) supplemented with encapsulated NO_3_^−^ at 2% and 3% in diet DM, respectively [9]. Lower rate of reduction (6.2%) was also reported for backgrounding steers fed a high-forage diet (65% DM corn silage) supplemented with encapsulated NO_3_^−^ (2.5% in diet DM [10]). Little is known about the factors that may interfere with the efficiency of NO_3_^−^ reduction in the rumen. Encapsulation of NO_3_^−^ [8], amount of NO_3_^−^ ingested, and intake rate of NO_3_^−^ [32,35], type of diet (e.g., roughage inclusion, N and S concentrations [10,11,35] as well as type of animals [35]) affect ruminal NO_3_^−^ utilization, and consequently, CH_4_ reduction. Furthermore, duration of feeding a dietary additive may affect its efficacy in reducing enteric CH_4_ production over time [36]; however, there was no decline in the effectiveness of EN over time in the current study (Figure 3). 

Multiple in vitro [8] and in vivo studies [4,5,31] have reported an increase in hydrogen production after feeding NO_3_^−^. Similarly, a significant (*p* < 0.001) increase in hydrogen production and yield was observed for the EN treatment in our study. It is generally believed that NO_3_^−^ reduction is a thermodynamically favorable process relative to methanogenesis in which NO_3_^−^ acts as a hydrogen sink [37]. However, considering the observed increase in hydrogen production following NO_3_^−^ supplementation in previous studies as well as in the current study, the earlier hypothesis needs to be re-examined. Perhaps the direct toxicity of NO_3_^−^ and its reduced intermediate (NO_2_^−^) on rumen microbes [3] may contribute as an additional mode of action in decreasing enteric CH_4_ production. Furthermore, hydrogen is an energy-dense gas (142 kJ/g of hydrogen, [38]) and its emission by animals could partially offset the energy gain by the decrease in CH_4_ production. For example, the calculated energy lost in hydrogen production for the +EN treatment was 23.3 kcal per day or 6.9% of the observed CH_4_ decrease with the use of EN. 

The increased concentration of plasma NO_3_^−^ and NO_2_^−^ following supplementation of NO_3_^−^ in the diets of ruminants implies that NO_3_^−^ is not fully reduced to ammonia. Lee et al. [10,32] observed a dose-response increase in blood NO_3_^−^ and NO_2_^−^ when encapsulated NO_3_^−^ was fed. The observed increase in blood NO_3_^−^ concentration for the EN treatment in our study was comparable to previous reports [10,33]. Furthermore, NO_2_^−^ was present in the blood in a detectable range (2 to 3 µg/L of NO_2_^−^-N) but did not elevate blood MetHb levels (less than 4.1% of total Hb) to the threshold that is considered to cause subclinical methemoglobinemia (30 to 40% [12]). Feeding NO_3_^−^ at 2 to 3% of dietary DM has been widely reported without any toxicity issues [30]. Although NO_3_^−^ consumption was relatively consistent over the experimental periods, blood NO_3_^−^ and NO_2_^−^ concentrations gradually decreased during the study, which could be due to a combination of factors, including a possible gradual improvement in microbial capacity to reduce dietary NO_3_^−^ [39], physiological change of the experimental animals [8], and change in the feeding behavior [8,10,40]. The lower feeding rate (g DM/min) and higher meal frequency per day for EN may have helped to spread out the availability of NO_3_^−^ to rumen microbes over a longer period, thus reducing the size of NO_3_^−^ pulses occurring in the rumen, which in turn would have lowered the concentration of NO_3_^−^ and NO_2_^−^ in rumen fluid and blood.

### 4.2. Essential Oils

The lack of effect of MBEO on average DMI and feed efficiency observed in our study is consistent with previous in vivo studies that supplemented EO or blend of EO to beef cattle. For example, Beauchemin and McGinn [41] reported that growing beef cattle fed a high-forge diet (75% DM whole-grain barley silage) supplemented with 1 g/day blend of EO (Crina^®^ Ruminants, mixture of thymol, eugenol, vanillin, limonene, and guaiacol) did not show any difference in DMI and feed efficiency. Similarly, using that same product (Crina^®^ Ruminants) at 1 and 2 g/day, Tomkins et al. [39] found no differences in DMI and animal performance for steers fed Rhodes grass hay (ad libitum). For beef animals on finishing diets, a study conducted by Yang et al. [20] tested the effects of 3 doses (0.4, 0.8, 1.6 g/day) of cinnamaldehyde or monensin on feedlot cattle performance, and reported that none of the treatments affected performance variables. Furthermore, Meyer et al. [42] reported no effect on DMI and feed efficiency for feedlot steers over a 115-day finishing period when fed a blend of EO (thymol, eugenol, vanillin, guaiacol, and limonene) at 1 g/animal/day. However, the authors reported improved efficiency for diets containing a blend of EO and tylosin. Furthermore, using higher doses (3.5 and 7 g/animal/day) of a blend of EO (MixOil^®^, extracts from oregano, garlic, lemon, rosemary, thyme, eucalyptus, and sweet orange), Rivaroli et al. [43] reported no effects on DMI and animal performance parameters for crossbred bulls fed high-grain finishing diets for 120-days. A similar lack of effect has been observed in other animal species, including sheep [44] and dairy cows [18,42]. Overall, the results from our study are consistent with the literature that suggests supplementation of diets with EO has no effects on DMI and performance of beef cattle. 

Only few in vivo studies investigated the effect of EO on enteric CH_4_ emission with conflicting results [16,17]. Furthermore, due to the variation in type of diets, dose rate, and the range of EO and EO compounds used, it is challenging to make a direct comparison of CH_4_ emission outputs among studies. Reduction in enteric CH_4_ emissions following supplementation of EO and blend of EO has been reported for sheep [23,44] and buffalo [45], although others reported no effect on CH_4_ emissions for beef [41,46] and dairy cattle [47]. However, in the current study, enteric CH_4_ emissions and yield were increased by 11.0% and 13.6%, respectively. It is difficult to explain the observed increase in CH_4_ production. However, several factors, including the wide range of non-specific antibacterial activity of EO that may favor methanogenesis [15] or positive impacts of EO on ruminal feed degradability [19,48], may play a role.

Information on the effect of EO on the process of methanogenesis in ruminants is ambiguous [49,50]. The impact of EO on CH_4_ emissions may be attributed to direct impact on methanogenic archaea (changing community structure or activity of methanogenesis pathway) and indirect impact on microbial metabolic processes contributing to methanogenesis [17]. Essential oils can also affect some protozoa that are symbiotically associated with archaea. Using meta-analysis, Khiaosa-ard and Zebeli [51] reported a dose-response effect of EO on reducing protozoa, whereas Cobellis et al. [17] reported no effect of EO on protozoa in most in vivo studies in ruminants. Furthermore, the antimicrobial activity of EO varies with the quantity used, chemical composition (both components present and their proportion), interaction among EO components, and chemical configurations [52,53]. Additive, antagonistic, and synergistic effects have been observed between components of EO [15,52]. The Activo^®^ Premium used in the MBEO treatment contained carvacol, eugenol, capsaicin, cinnamaldehyde, and pepper extract with diverse antimicrobial activities. For example, carvacrol has shown a negative effect on Gram-negative bacteria [54], whereas eugenol has shown a broad antibacterial activity by affecting both Gram-negative and Gram-positive bacteria [14,55]. 

Although reduction in DM digestibility following supplementation of EO has been reported in most in vitro studies, in vivo studies have been inconsistent [17]. Yang et al. [48] fed garlic (5 g/cow per day) and juniper berry (2 g/cow per day) to dairy cows consuming a ration containing forage (40% DM) and reported a 12 to 15% increase in rumen DM and OM digestibility. However, total tract digestibility of DM, OM, fiber, and starch were not affected. They suggested that the increased ruminal digestibility was due to an 11% increase in dietary protein digestibility in the rumen compared with control. In another study, Silva et al. [19] fed Activo^®^ Premium at the rate of 150 mg/kg DM to dairy cows consuming a diet containing corn silage (48% DM) and observed an increase in total tract OM digestibility. It has also been reported that the effects of EO and EO blends are rumen pH and diet dependent [14]. Benchaar et al. [56] reported that supplementation of 750 mg Crina^®^ Ruminants per day to dairy cows tended to increase total VFA concentration in the rumen of lactating cows when the diet contained alfalfa silage, but tended to decrease total VFA concentration when the diet contained corn silage. Overall, further long-term in vivo studies are required to determine the potential of using EO in ruminant diets to lower enteric CH_4_ production.

Although studies report anti-inflammatory, immune modulation, antioxidant, thermoregulation, and blood oxygenation properties of EO [14,18], blood parameters measured in our study did not differ for the MBEO and EN + MBEO treatments. Recent findings suggest that due to their phenolic nature, some EO are likely less susceptible to microbial degradation in the rumen and exhibit activities post-ruminally by binding to specific receptors expressed in neurons, intestines, and other cells [18,57]. However, Oh et al. [18] stated that these impacts of EO are likely dependent on the type and physiological status of the experimental animals, as well as the type of diets. In dairy cattle, providing EO either in the diet or by direct infusion into the abomasum had an effect on the immune system of the animals, post-ruminal nutrient use, and animal physiology [18]. Silva et al. [19,22] fed Activo^®^ Premium at 150 mg/kg diet DM to dairy cows in mid-lactation for 8 weeks and reported increased O_2_ saturation of Hb and a greater proportion of O_2_ transported by blood in relation to total gases for cows fed a blend of EO compared with a control. Furthermore, for feedlot cattle fed a finishing diet containing dry-rolled barley grain (86% DM), Yang et al. [20] reported that supplementation with cinnamaldehyde (0.4 to 1.6 g/day per animals) reduced stress and increased DMI during the early feeding period when stress is greater. In our study, animals were fed a high-forage diet (80% DM) and were likely under minimal stress. 

## 5. Conclusions

The effects of feeding EN as a replacement for urea and MBEO alone or in combination with EN on animal performance and enteric CH_4_ emissions from beef steers fed a high-forage diet were investigated. Our results demonstrate that there were no advantages of feeding EN with MBEO. Supplementing diets with MBEO neither improved animal performance nor lowered CH_4_ emissions. However, EN reduced CH_4_ emissions and altered feeding behavior, whereas it had no impact on animal health and performance. Accordingly, the use of EN could have important implications for the Canadian beef sector in particular and global ruminant agriculture in general. In 2016, CH_4_ emissions from enteric fermentation represented 3.5% of the Canadian national greenhouse gas inventory, with beef cattle production accounting for 80% of the emissions. A 30% adoption rate of EN by beef producers, combined with 17.6% reduction of enteric CH_4_ emissions following the use of EN, would result in 4% less total enteric emissions and 1.8% less agricultural emissions in Canada. 

## Figures and Tables

**Figure 1 animals-09-00021-f001:**
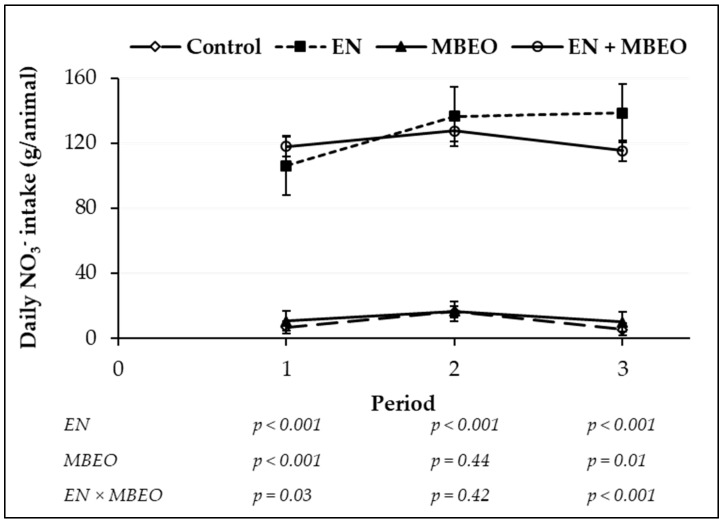
Nitrate intake of backgrounding beef steers fed a high-forage diet with no additives (control, −EN −MBEO) or supplemented with encapsulated nitrate (EN), microencapsulated blend of essential oils (MBEO), and combination of EN and MBEO (EN + MBEO). Error bars indicate standard deviation. For the statistical analysis, EN represents the main effect of encapsulated nitrate (−EN −MBEO and −EN +MBEO) versus (+EN −MBEO and +EN +MBEO); MBEO represents the main effects of microencapsulated blend of essential oils (−EN −MBEO and +EN −MBEO) versus (−EN +MBEO and +EN +MBEO); EN + MBEO represents the interaction between main effects of EN and MBEO.

**Figure 2 animals-09-00021-f002:**
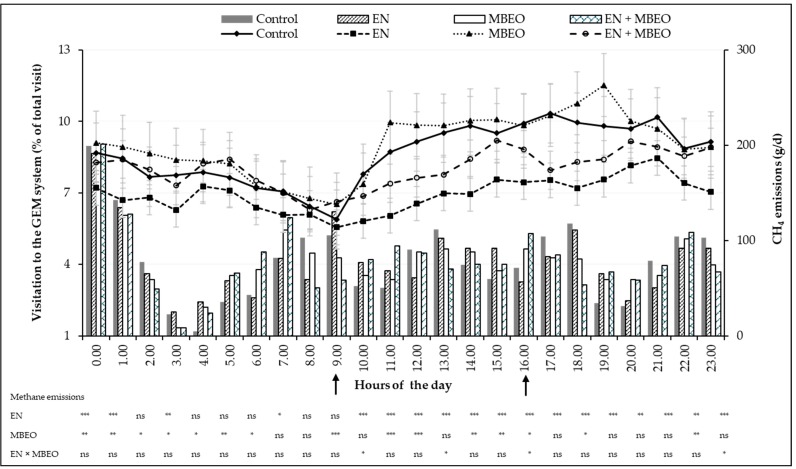
Diurnal pattern of CH_4_ emissions (g/day, solid and broken lines) and animal visits to the GreenFeed emission monitoring (GEM) system over 24-h period (% of total visits, bar graphs) for backgrounding beef steers fed a high-forage diet with no additives (control, −EN, −MBEO), or supplemented with encapsulated nitrate (EN), microencapsulated blend of essential oils (MBEO), and combination of EN and MBEO (EN + MBEO). The arrows indicate time of feeding at 0900 h and 1600 h, and 0000 h indicates midnight. There were no significant differences among treatments for the animal visits at individual time points throughout the hours of the day. For CH_4_ emissions, error bars indicate standard deviation and ns = *p* > 0.05; * *p* ≤ 0.05; ** *p* ≤ 0.01; *** *p* ≤ 0.001. For the statistical analysis, EN represents the main effect of encapsulated nitrate (−EN −MBEO and −EN +MBEO) versus (+EN −MBEO and +EN +MBEO); MBEO represents the main effects of microencapsulated blend of essential oils (−EN −MBEO and +EN −MBEO) versus (−EN +MBEO and +EN +MBEO); EN + MBEO represents the interaction between main effects of EN and MBEO.

**Figure 3 animals-09-00021-f003:**
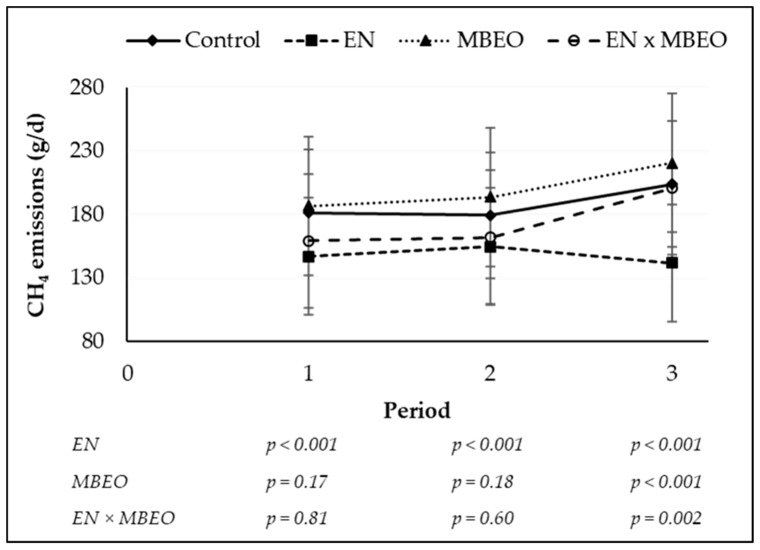
Enteric CH_4_ emissions over the experimental period for beef steers consuming a high-forage diet with no additives (control, −EN −MBEO) or supplemented with encapsulated nitrate (EN), microencapsulated blend essential oils (MBEO), and combination of EN and MBEO (EN + MBEO). Error bars indicate standard deviation. For the statistical analysis, EN represents the main effect of encapsulated nitrate (−EN −MBEO and −EN +MBEO) versus (+EN −MBEO and +EN +MBEO); MBEO represents the main effects of microencapsulated blend of essential oils (−EN −MBEO and +EN −MBEO) versus (−EN +MBEO and +EN +MBEO); EN + MBEO represents the interaction between main effects of EN and MBEO.

**Figure 4 animals-09-00021-f004:**
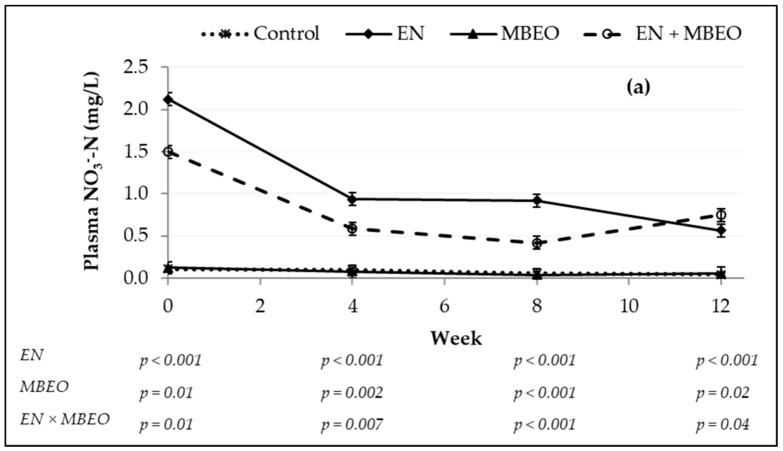
Plasma NO_3_^−^-N (**a**) and NO_2_^−^-N (**b**) concentration before morning feeding in beef steers (*n* = 22) consuming a high-forage diet with no additives (control, −EN, −MBEO) or supplemented with encapsulated nitrate (EN), microencapsulated blend essential oils (MBEO), and combination of EN and MBEO (EN + MBEO). Week zero indicates the end of adaptation period (experimental day 28). Error bars indicate standard deviation. For the statistical analysis, EN represents the main effect of encapsulated nitrate (−EN −MBEO and −EN +MBEO) versus (+EN −MBEO and +EN +MBEO); MBEO represents the main effects of microencapsulated blend of essential oils (−EN −MBEO and +EN −MBEO) versus (−EN +MBEO and +EN +MBEO); EN + MBEO represents the interaction between main effects of EN and MBEO.

**Table 1 animals-09-00021-t001:** Feed ingredients and chemical composition of the experimental diets with no additives (control, −EN, and −MBEO), or supplemented with encapsulated nitrate (+EN), microencapsulated blend of essential oils (+MBEO), and combination of EN and MBEO (+EN +MBEO).

Item	−EN ^1^		+EN	
	−MBEO ^1^	+MBEO	−MBEO	+MBEO
Ingredients, % of dry matter (DM)				
Corn silage ^2^	80	80	80	80
Barley grain, dry rolled ^3^	10	10	10	10
Supplement	10	9.99	7.5	7.48
Canola meal	3.70	3.70	3.70	3.70
Limestone	1.55	1.55	0.43	0.43
Salt (NaCl)	0.11	0.11	0.11	0.11
Urea	1.17	1.17	0.24	0.24
LeRDC beef feedlot premix ^4^	0.05	0.05	0.05	0.05
Molasses, dried	0.05	0.05	0.05	0.05
Barley ground	3.31	3.31	2.85	2.85
Canola oil	0.07	0.05	0.07	0.06
EN	0.0	0.0	2.5	2.5
MBEO	0.0	0.015	0.0	0.015
Chemical composition (% of DM)				
DM (as-is)	44.0	44.0	43.8	43.8
OM	93.9	93.9	92.5	92.5
CP	14.3	14.3	13.1	13.1
NDF	42.0	42.0	41.2	41.1
ADF	28.1	28.1	27.9	27.9
Starch	29.8	29.8	29.3	29.3
NO_3_^−^	0.12	0.16	1.66	1.68
GE (Mcal/kg DM) ^5^	5.39	5.39	5.32	5.29

DM = dry matter; OM = organic matter; CP = crude protein; NDF = neutral detergent fiber; ADF = acid detergent fiber; NO_3_^−^ = nitrate. ^1^ EN = Encapsulated nitrate (EN) was manufactured by GRASP Ind. & Com. LTDA, Curitiba, Paraná, Brazil; DM, 85.6%; N, 17.6%; Ca, 19.6%; and NO_3_^−^, 71.4% on a DM basis. The source of nitrate was the double salt of calcium ammonium nitrate decahydrate [5Ca(NO_3_)_2_•NH_4_NO_3_•10H_2_O]; MBEO = Commercial microencapsulated blend of natural and identical to natural terpenoids (carvacrol), phenylpropanoids (cinnamaldehyde and eugenol), and alkaloids (capsaicin from capsicum oleoresin) manufactured by GRASP Ind. & Com. LTDA, Curitiba, Paraná, Brazil. ^2^ DM, 32% (±1.3 SD) on as-is basis and OM, 96% (±0.1 SD); CP, 8% (±0.2 SD); NDF, 47.3% (±2.1 SD); ADF, 32.8% (±3.8 SD); starch, 28% (±3.6 SD) on a DM basis. ^3^ OM, 98% (±0.5 SD); CP, 14% (±0.8 SD); NDF, 16.6% (±1.8 SD); starch, 55% (±1.6 SD) on a DM basis. ^4^ Lethbridge Research and Development Centre (LeRDC) beef feedlot vitamin-mineral premix contained (on a DM basis) CaCO_3_, 34.83%; ZnSO_4_, 28.37%; CuSO_4_, 10.31%; ethylenediamine dihydriodide (80% concentration), 0.15%; selenium 1% (10,000 mg Se/kg, Na_2_SeO_3_), 5.04%; CoCO_3_, 0.08%; MnSO_4_, 14.61%; vitamin A (500,000,000 IU/kg), 1.72%; vitamin D (500,000,000 IU/kg), 0.17%; and vitamin E (500,000 IU/kg), 4.73%. ^5^ Gross energy (GE, Mcal/kg DM); corn silage, 5.60 (±0.35 SD); dry rolled barley grain, 4.91 (±0.05 SD); control and MBEO supplement, 4.24 (±0.15 SD); and EN supplement, 4.69 (±0.18 SD).

**Table 2 animals-09-00021-t002:** Body weight, average daily gain, dry matter intake, feed efficiency, and feeding behavior of backgrounding beef steers fed a high-forage diet with no additives (control, −EN −MBEO), or supplemented with encapsulated nitrate (+EN), microencapsulated blend of essential oils (+MBEO), and combination of EN and MBEO (+EN +MBEO).

Item ^2^	−EN ^1^		+EN ^1^		SEM	*p*-Value			
	−MBEO	+MBEO	−MBEO	+MBEO		EN	MBEO	EN × MBEO	Period
Number of animals	22	22	22	22	---	---	---	---	---
**Body weight, kg**									
Initial	332	331	331	331	5.03	0.939	0.953	0.989	---
day 28	371	363	359	356	5.59	0.088	0.348	0.604	---
day 56	397	395	387	382	6.12	0.060	0.541	0.839	---
day 84	439	426	413	407	6.92	0.002	0.184	0.561	---
Final (d 112)	467	458	446	441	7.60	0.012	0.336	0.835	---
**ADG (kg/day)**									
day 1 to 28	1.417	1.137	0.986	0.910	0.060	<0.0001	0.004	0.093	---
day 29 to 56	0.934 ^b^	1.136 ^a^	1.010 ^a,b^	0.916 ^b^	0.050	0.153	0.281	0.004	---
day 57 to 84	1.497 ^a^	1.110 ^b^	0.917 ^b^	0.909 ^b^	0.055	<0.0001	0.001	0.001	---
day 85 to 112	0.998	1.153	1.190	1.173	0.053	0.049	0.200	0.111	---
**DMI (kg/day) ^3^**									
day 1 to 28	7.71	7.53	7.05	7.12	0.180	0.004	0.768	0.489	---
day 29 to 56	7.85	8.31	7.82	7.69	0.180	0.072	0.359	0.100	---
day 57 to 84	9.35	8.81	8.67	8.47	0.197	0.011	0.063	0.388	---
day 85 to 112	9.47	9.53	8.95	8.73	0.180	0.001	0.673	0.447	---
**G:F**									
day 1 to 28	0.184	0.150	0.139	0.127	0.007	<0.0001	0.002	0.123	---
day 29 to 56	0.118 ^b^	0.136 ^a^	0.129 ^a,b^	0.118 ^b^	0.006	0.504	0.571	0.009	---
day 57 to 84	0.160 ^a^	0.126 ^b^	0.106 ^c^	0.105 ^c^	0.005	<0.0001	0.002	0.002	---
day 85 to 112	0.105	0.121	0.132	0.134	0.005	0.0001	0.088	0.182	---
**Average (day 29 to 112)**									
DMI (kg/day) ^3^	8.93	8.87	8.47	8.28	0.176	0.003	0.479	0.713	<0.0001
ADG (kg/day)	1.065	1.157	1.073	1.018	0.042	0.119	0.657	0.080	0.0001
G:F	0.123	0.128	0.125	0.121	0.004	0.432	0.921	0.240	0.811
Average daily NO_3_^–^ consumed (g/animal) ^4^	9.43	12.43	127.20	120.37	5.030	<0.001	0.723	0.370	0.160
**Feeding behavior ^5^**									
Total meal duration, min/day	183.6	183.4	188.2	186.6	4.71	0.410	0.849	0.883	<0.0001
Head down duration per meal, min/meal	9.6	9.3	10.2	9.6	0.68	0.572	0.519	0.793	0.323
Head down duration, min/day	80.9	84.0	97.1	91.6	4.83	0.016	0.810	0.377	<0.0001
Feeding/meal frequency, events/day	9.2	9.6	10.4	10.4	0.30	0.002	0.495	0.440	<0.0001
Feeding rate, g DM/min	46.2	46.1	42.7	41.1	1.29	0.001	0.514	0.553	<0.0001

^1^ EN represents the main effect of encapsulated nitrate (−EN −MBEO and −EN +MBEO) versus (+EN −MBEO and +EN +MBEO); MBEO represents the main effects of microencapsulated blend of essential oils (−EN −MBEO and +EN −MBEO) versus (−EN +MBEO and +EN +MBEO); +EN +MBEO represents the interaction between main effects of EN and MBEO. ^2^ Animals that received diets containing encapsulated NO_3_^−^ were acclimatized gradually using a step-up protocol during the first 28 days of adaptation; 0.625%, 1.25%, 1.875%, and 2.5% NO_3_^−^ in dietary DM. Animals that received MBEO were supplemented with 150 mg/kg DM microencapsulated blend of EO since the beginning of the experiment. ^3^ Dry matter intake for animals that visited the GreenFeed emission monitoring system included the amount of pellet consumed while visiting the system. ^4^ NO_3_^−^ consumption was calculated from NO_3_^−^ analysis of TMR and ort samples. ^5^ A meal was defined as a visit to the bunk, followed by an absence from the bunk for 300 s or greater. Total meal duration = total time spent at feeder, head down duration per meal = meal duration/number of meals per day, feeding rate = DM consumed by time at feeder (DMI/meal duration). ^a, b, c^ Means within a row for each treatment with different lower case letter are significantly different (*p* ≤ 0.05).

**Table 3 animals-09-00021-t003:** Particle size distribution of total mixed ration (TMR) and orts (*n* = 3) from beef steers fed a backgrounding diet with no additives (control, −EN −MBEO), or supplemented with encapsulated nitrate (+EN), microencapsulated blend of essential oils (+MBEO), and combination of EN and MBEO (+EN +MBEO).

Item	−EN ^1^		+EN ^1^		SEM	*p*-Value			
	−MBEO	+MBEO	−MBEO	+MBEO		EN	MBEO	EN × MBEO	Period
**TMR, % (as-is basis)**									
Large (≥18 mm)	2.57	3.93	3.43	3.23	0.331	0.817	0.128	0.056	0.020
Medium (8 to 18 mm)	61.58	66.65	65.19	66.61	1.917	0.462	0.219	0.451	0.413
Small (1.2 to 8 mm)	31.99	28.13	29.63	28.89	1.430	0.706	0.388	0.510	0.707
Bottom (<1.2 mm)	2.92	1.83	1.41	2.02	0.554	0.479	0.766	0.401	0.269
**Orts, % (as-is basis)**									
Large (≥18 mm)	11.48	14.27	12.65	9.52	3.886	0.732	0.973	0.586	0.115
Medium (8 to 18 mm)	57.26	62.71	61.01	58.86	2.211	0.982	0.483	0.136	0.111
Small (1.2 to 8 mm)	29.65 ^a^	24.11 ^b^	24.02 ^b^	30.70 ^a^	0.775	0.564	0.49	0.0001	0.001
Bottom (<1.2 mm)	1.71 ^a^	0.79 ^a,b^	0.49 ^b^	0.78 ^a,b^	0.232	0.039	0.225	0.04	0.017
Orts, % of total offered (DM basis)	1.08	1.01	1.36	2.62	0.477	0.096	0.259	0.215	0.037

^1^ EN represents the main effect of encapsulated nitrate (−EN −MBEO and −EN +MBEO) versus (+EN −MBEO and +EN +MBEO); MBEO represents the main effects of microencapsulated blend of essential oils (−EN −MBEO and +EN −MBEO) versus (−EN −MBEO and +EN +MBEO); +EN +MBEO represents the interaction between main effects of EN and MBEO. ^a, b^ Means within a row for each treatment with different lower case letter are significantly different (*p* ≤ 0.05).

**Table 4 animals-09-00021-t004:** Visitation to the GreenFeed emissions monitoring (GEM) system, pellet consumption, and emission and yield of CH_4_ and hydrogen from backgrounding beef animals fed a high-forage diet with no additives (control, −EN, −MBEO) or supplemented with encapsulated nitrate (+EN), microencapsulated blend of essential oils (+MBEO), and combination of EN and MBEO (+EN +MBEO).

Item	−EN ^1^		+EN ^1^		SEM	*p*-Value			
	−MBEO	+MBEO	−MBEO	+MBEO		EN	MBEO	EN × MBEO	Period
**GEM system visitation**									
Number of animals that visited	18 ^a^	16 ^b^	17 ^a,b^	18 ^a^	0.5	0.741	0.339	0.021	0.010
Good visits per animal per period ^2^	33.0	33.1	33.3	34.3	1.66	0.647	0.743	0.807	0.210
Visit duration (min:s)	4:06	4:07	4:39	4:18	0.12	0.089	0.210	0.389	<0.0001
Pellet consumed, kg DM/day	0.79	0.84	0.82	0.88	0.030	0.262	0.065	0.911	<0.0001
DMI ^3^, kg/day	9.02	8.89	8.56	8.30	0.207	0.010	0.322	0.713	<0.0001
GE intake, Mcal/day	46.64	47.30	44.83	43.56	1.075	0.009	0.766	0.353	<0.0001
**CH_4_**									
g/day	184.14	197.69	145.89	168.78	6.533	<0.0001	0.005	0.458	<0.0001
g/kg of DMI	20.69	22.35	17.00	20.46	0.683	<0.0001	0.0002	0.174	<0.0001
% of GE intake	5.19	5.55	4.46	5.12	0.169	0.001	0.002	0.350	0.0002
**Hydrogen ^4^**									
g/day	0.428 ^c^	0.455 ^c^	0.734 ^a^	0.639 ^b^	0.0265	<0.0001	0.192	0.022	<0.0001
g/kg of DMI	0.050	0.053	0.084	0.078	0.0031	<0.0001	0.571	0.123	<0.0001

^1^ EN represents the main effect of encapsulated nitrate (−EN −MBEO and −EN +MBEO) versus (+EN −MBEO and +EN +MBEO); MBEO represents the main effects of microencapsulated blend of essential oils (−EN −MBEO and +EN −MBEO) versus (−EN +MBEO and +EN +MBEO); +EN +MBEO represents the interaction between main effects of EN and MBEO. ^2^ Good visits were selected based on the distance of the animal’s head from the proximity sensor and the duration that the animal’s head in the “head chamber”. Good visits were used to calculate the average daily CH_4_ emissions. ^3^ DM intake included both TMR and pellet consumption. ^4^ Hydrogen emission was calculated using the “arithmetic averaging method”, a straight-forward averaging of the visit fluxes defined as the sum of the visit fluxes divided by the number of measurements [26]. ^a, b, c^ Means within a row for each treatment with different lower case letter are significantly different (*p* ≤ 0.05).

**Table 5 animals-09-00021-t005:** Jugular blood acid-base balance of beef steers (*n* = 22) fed a high forage backgrounding diet with no additives (control, −EN, −MBEO) or supplemented with encapsulated nitrate (+EN), microencapsulated blend of essential oils (+MBEO), and combination of EN and MBEO (+EN +MBEO).

Item ^2^	−EN ^1^		+EN ^1^		SEM	*p*-Value			
	−MBEO	+MBEO	−MBEO	+MBEO		EN	MBEO	EN × MBEO	Period
pCO_2_, mmHg	40.91	39.55	41.39	41.82	0.974	0.165	0.632	0.364	<0.0001
tCO_2_, mmol/L ^3^	29.30	29.55	30.43	30.49	0.506	0.048	0.762	0.844	<0.0001
HCO_3_^−^, mmol/L ^3^	28.06	28.34	29.15	29.20	0.486	0.051	0.724	0.813	<0.0001
BE, mmol/L ^3^	4.82	5.28	5.49	5.65	0.380	0.180	0.424	0.691	0.209
pH	7.48	7.50	7.48	7.49	0.008	0.857	0.334	0.200	0.0001
pO_2_, mmHg	42.79	43.03	41.88	44.03	1.127	0.968	0.294	0.402	0.0077
SatO_2_, % total Hb	78.00	79.15	77.27	78.91	1.223	0.694	0.261	0.844	<0.001
PCV, %	43.03	43.40	43.49	43.83	0.571	0.444	0.539	0.979	0.0001
Total Hb, g/dL	16.15	15.98	16.42	16.29	0.210	0.170	0.479	0.934	<0.0001
MetHb, g/100 g Hb	0.70 ^b^	0.87 ^b^	1.45 ^a^	1.22 ^a^	0.053	<0.0001	0.710	0.008	0.0003
Min., g/100 g Hb	0.00	0.25	0.20	0.55	---	---	---	---	---
Max., g/100 g Hb	1.95	2.53	3.40	4.10	---	---	---	---	---
Plasma									
NO_3_^−^-N, mg/L	0.082 ^c^	0.076 ^c^	1.135 ^a^	0.812 ^b^	0.052	<0.0001	0.002	0.003	<0.0001
NO_2_^−^-N, µg/L	2.536	2.621	2.178	1.971	0.129	0.0002	0.636	0.260	<0.0001

^1^ EN represents the main effect of encapsulated nitrate (−EN −MBEO and −EN +MBEO) versus (+EN −MBEO and +EN +MBEO); MBEO represents the main effects of microencapsulated blend of essential oils (−EN −MBEO and +EN −MBEO) versus (−EN +MBEO and +EN +MBEO); +EN +MBEO represents the interaction between main effects of EN and MBEO. ^2^ pCO_2_, partial pressure of carbon dioxide; tCO_2_, total concentration of CO_2_; HCO_3_^−^, bicarbonate; BE, base excess; pO_2_, partial pressure of O_2_; SatO_2_, O_2_ saturation as percent of oxygen based on total hemoglobin saturation capacity; PCV, packed cell volume; Hb, hemoglobin; MetHb, methemoglobin; NO_3_^−^, nitrate and NO_2_^−^, nitrite. ^3^ These parameters were calculated from parameters measured by VetStat analyzer. ^a, b, c^ Means within a row for each treatment with different lower case letter are significantly different (*p* ≤ 0.05).

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
