# Peer review of "Effect of Encapsulated Nitrate and Microencapsulated Blend of Essential Oils on Growth Performance and Methane Emissions from Beef Steers Fed Backgrounding Diets"

_animals, 2019, doi:10.3390/ani9010021_

Round 1
Reviewer 1 Report
The paper explores the effects of encapsulated NO3 and microencapsulated EO, and their interaction, on performances and CH4 emissions in beef cattle. The experiment is complete and well done. It is worthy to note that “Introduction” section is a perfect example of scientific writing: simple, concise, clear and complete. The results are perfectly aligned whit the literature cited and correctly interpreted. However, there are some statistics that must be better explained to improve the readability of this good paper. In particular, I do not understand why two data that in several figures are superimposed are declared different. For example, in in Fig. 3, Control and EMxMBEO that seem to be very close (if not the same) in the graph are declared different (P=0.02). The same in Fig. 4 in which Control and MBEO are superimposed (or seem to be) for the whole experimental period. Perhaps SE is very small, but this not seems this case looking at the large SD. The conclusions are fairly dry: the authors shot a brief statement, to answer the hypothesis, reporting no effects of treatments on the phenomena they are looking at. However, I think that the reduction of 13% (about 1/8) of CH4 emissions is good news, especially if coupled with no change in animal performances! Because LCAs in beef cattle indicate CH4 as responsible of half of total emissions, it means that global carbon footprint can be reduced by 6-7% by using EN in field whit the feeding protocol utilised by the authors.
Here some punctual observations.
Row 62. Citation [12] not pertinent
Row 114 Diets are not isonitrogenous (13.1% vs 14.7% of CP on DM)
Rows 375-7 Citation
Row 384 6.9% is a part of CH4 or contribute to the total amount?
Rows 418-9 Not affect
Rows 419-21 Delete: further studies are always needed!
Rows 482-4 The same.
Reviewer 2 Report
Line 76: thermoregulation
Line 99: The four treatments (Table 1)... Remove reference to diets from results.
Line 135: all animals
Line 142: manufacturer and location of company for GEM
